# RETHINKING INDEPENDENT CROSS-ENTROPY LOSS FOR GRAPH-STRUCTURED DATA

## ABSTRACT

Graph neural networks (GNNs) have exhibited prominent performance in learning graph-structured data. Considering node classification task, the individual label distribution conditioned on node representation is used to predict its classes. Based on the i.i.d assumption among node labels, the traditional supervised learning simply sums up cross-entropy losses of the independent training nodes and applies the average loss to optimize GNNs' weights. But different from other data formats, the nodes are naturally connected and their classes are correlated to neighbors at the same cluster. It is found that the independent distribution modeling of node labels restricts GNNs' capability to generalize over the entire graph and defend adversarial attacks. In this work, we propose a new framework, termed joint-cluster supervised learning, to model the joint distribution of each node with its corresponding cluster. Rather than assuming the node labels are independent, we learn the joint distribution of node and cluster labels conditioned on their representations, and train GNNs with the obtained joint loss. In this way, the data-label reference signals extracted from the local cluster explicitly strengthen the discrimination ability on the target node. The extensive experiments on 12 benchmark datasets and 7 backbone models demonstrate that our joint-cluster supervised learning can effectively bolster GNNs' node classification accuracy. Furthermore, being benefited from the reference signals which may be free from spiteful interference, our learning paradigm significantly protects the node classification from being affected by the adversarial attack. The code is available at: https://anonymous.4open.science/r/Joint-cluster-loss-01C7.

## 1 INTRODUCTION

Graph-structured data is ubiquitous in a broad spectrum of application domains, such as social networks (Perozzi et al., 2014; Fan et al., 2019), biological networks (Diao et al., 2022; Yu et al., 2023), recommender system (He et al., 2020; Wang et al., 2019), and knowledge graphs (Wang et al., 2018; Arora, 2020). Graph neural networks (GNNs) have been extensively explored to learn the complex connectivity information and node features in an end-to-end manner. Particularly, GNNs follow a message passing strategy and learn the representation vector of each node by iteratively aggregating the representations of its neighbors and combining with itself. The learned node representations facilitate various downstream tasks including node classification (Gasteiger et al., 2018; Zhou et al., 2020) and link prediction (Grover & Leskovec, 2016; Zhang & Chen, 2018).

Despite the persistent efforts in feature learning, label dependencies among nodes receives inadequate attentions. Considering the node classification task with GNNs, decision making is modeled by independent conditional distribution $P(y_i|z_i)$, where $y_i$ and $z_i$ are the label and learned feature of a specific node and its cross-entropy loss is $\text{CE}(y_i, P(y_i|z_i))$. However, it is notorious that such independent decision making of node label exacerbates following issues. *Overfitting*: The overly minimization of cross-entropy loss prefers the higher prediction probabilities (i.e., over-confident decision) on the small set of training nodes (Guo et al., 2017), resulting in poor generalization on the rest of graph(Guo et al., 2017). *Susceptibility to adversarial attacks*: The over-confident GNNs underestimate their uncertainties, which is often leveraged adversarial attacker to craft input examples that lie in uncertain regions but have different labels (Szegedy et al., 2013). This presents a challenge to calibrate GNNs' training and hence generate robust decision making.

On the other hand, the decision making based on i.i.d assumption of node label is not in line with the graph-structured data, where nodes tend to connect with "similar" neighbors to form some clusters. The i.i.d assumption factorizes the joint distribution into a product of multiple prediction densities: $P(y_1, \cdots, y_n | z_1, \cdots, z_n) = \prod P(y_i | z_i)$. This straightforward factorization fails to comprehensively account for the inherent node correlations. Although the message passing learns the neighborhood-aware node features, the label dependencies are threw out during node inference. Just like human experts making decisions with other data-label pairs as reference signals, GNN models could promote their reasoning capabilities via the prompt data. We are aware of the previous arts in investigating the label dependencies (Huang et al., 2020; Ma et al., 2018); but they either cannot unite with the feature learning in GNNs or have poor efficiency. In view of such, we ask:

*How we efficiently learn the joint distribution together with GNNs to reason node label rationally?*

In this paper, we propose a new framework named joint-cluster supervised learning to model the joint distribution of each node and its corresponding cluster. For an individual node, we learn the joint-cluster distribution of $P(y_i, y_c | z_i, z_c)$, where $z_c$ and $y_c$ are the constructed cluster feature and label, respectively. The motivation for adopting cluster is to provide the sufficient reference signals for target sample, while reducing computational complexity required to integrate the remaining nodes in the vanilla joint distribution. Particularly, we optimize GNNs by minimizing the joint-cluster cross-entropy loss. The well-trained GNNs are then leveraged to infer the node label by marginalizing the joint-cluster distribution as shown in Figure 1. *Compared to supervised learning, the main difference of our work is to explicitly learn the joint density of the target sample and its reference signals.* The contributions are summarized below:

- We introduce a new paradigm of joint-cluster supervised learning for graph data. By breaking the i.i.d assumption in node classes and loss computation, we propose to model the joint distribution between the target node and its located cluster, and leverage it to train and infer GNNs.

- The joint distribution disperses prediction densities of a node over a larger label space, thereby relieving the over-confident decision making. We comprehensively test on small, large, class-imbalanced, and heterophilic graphs. The experiments on 12 datasets and 7 backbone models consistently validate the substantial generalization capability of joint-cluster distribution learning.

- The joint-cluster distribution learning generates more robust classifications for the attacked nodes compared with the independent decision making, owing to the reliable reference signal of cluster.

- The joint-cluster supervised learning surpasses the state-of-the-art (SOTA) models that encode the label dependencies, in terms of the node classification accuracy, training and inference efficiencies.

## 2 PRELIMINARY OF GNNS AND SUPERVISED LEARNING

We focus on the node classification task to introduce the new concept of joint distribution learning. Let $(x_i, y_i)$ denote the node-label pair, where $x_i \in \mathbb{R}^d$ and $y_i \in \mathbb{R}^c$ are input feature and label vector of node $v_i$, resepctively, $d$ and $c$ are the dimension sizes. Given training set $\{(x_i, y_i)\}_{i=1}^{L}$, where $L$ is the number of labeled nodes, the goal of node classification task is to train a predictor $f_\theta : \mathbb{R}^d \rightarrow \mathbb{R}^c$, mapping each node over the entire graph to a desired label with trainable parameter $\theta$.

### 2.1 GRAPH NEURAL NETWORKS

GNNs have emerged as one of the standard tools to learn both the node features and graph structure. Mathematically, based on the recursive message passing mechanism, at the $k$-th layer of GNNs, the embedding vector $z_i^{(k)}$ of each node $v_i$ is obtained by (Xu et al., 2018):

$$z_i^{(k)} = \text{Aggregate}(\{z_j^{(k-1)} \mid \forall j \in \mathcal{N}(i) \cup i\}; \theta). \tag{1}$$

Function $\text{Aggregate}$ denotes combination operator (e.g., sum, mean, or max) on the neighborhood embeddings, and $\mathcal{N}(i)$ denotes a set of neighbors connected to node $v_i$. Suppose we have a number of repeated message-passing layers. We simply use $z_i = f_\theta(x_i)$ to denote the final generated node representation of node $v_i$ and utilize it to predict the corresponding node label $\hat{y}_i$.

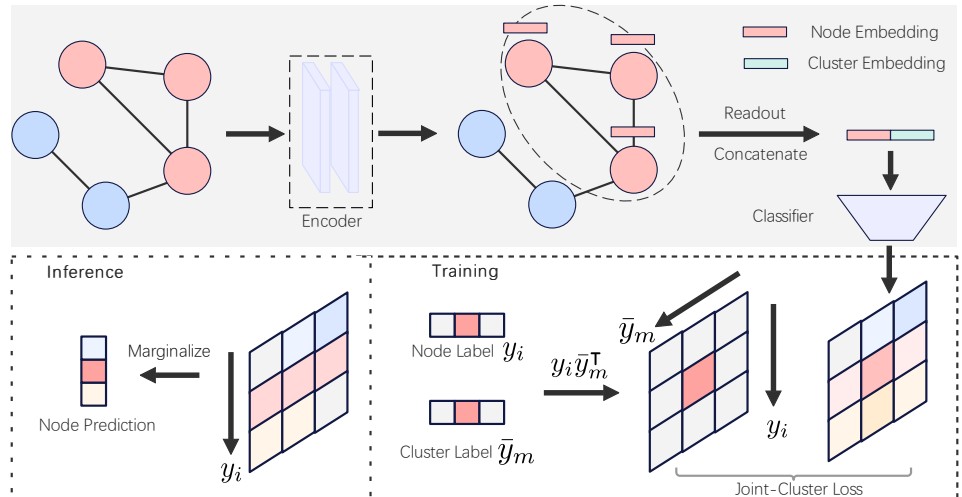

Figure 1: An illustration of our joint-cluster supervised learning framework: First, we obtain node embeddings through the encoder. Then the cluster embedding and label are generated through the divided graph structure. Then the node embedding and the cluster embedding are concatenated and fed into the classifier to obtain a joint distribution prediction. Finally, the joint-cluster loss and marginalization are used for training and inference.

## 2.2 INDEPENDENT CROSS-ENTROPY LOSS

Following the supervised learning paradigm and considering the training nodes, vanilla cross-entropy loss is obtained by $\mathcal{L}_{CE} = -\sum_{i=1}^{L} y_i \log \hat{y}_i$. This approach has been applied to multiple domains such as CV and NLP. Particularly, the supervised learning makes use of the conditional density $p(y \mid z)$ for each pair $(z_i, y_i)$ and train model weights via maximum likelihood estimator (MLE):

$$\hat{\theta}_{\text{CE}}\left(\{z_i, y_i\}_{i=1}^{L}\right) = \arg\max_{\theta} p(y_1, \ldots, y_L \mid z_1, \ldots, z_L; \theta) \tag{2a}$$

$$= \arg\max_{\theta} \prod_{i=1}^{L} p(y_i \mid z_i; \theta) = \arg\max_{\theta} \sum_{i=1}^{L} \log p(y_i \mid z_i; \theta). \tag{2b}$$

Note that we use $p(y_i \mid z_i; \theta)$ and $p(y_i \mid z_i)$ interactively, where the former one is used in the context of model optimization and the later one is adopted for simpleness. Eq.(2b) is deduced not according to mathematical consequence but based on the i.i.d assumption between nodes' labels. However, such decomposition is not desired in graph data, since the node features and classes are inherently correlated depending on the graph connectivity. Although GNNs aggregate the neighborhoods and make decision on the target node conditioned on the set of neighbors' features, the joint-distribution modeling of node classes is still broken in Eq.(2b). The prior knowledge that the nodes at the same cluster share similar labels is widely accepted in many real-world graphs, like social networks. These intuitions inspire us to learn the joint distribution of node classes conditioned on their features.

## 3 JOINT-CLUSTER SUPERVISED LEARNING

As analyzed before, GNNs utilize graph structure through the unique message passing, while still treating the node labels are independent from each other during the loss optimization. Despite the conceptual simpleness, it is not trivial to model the joint distribution. Given a number of training nodes, the fully joint conditional distribution $p(y_1, \ldots, y_L \mid z_1, \ldots, z_L; \theta)$ has to be constructed over label state space of $\mathbb{R}^{c^L}$. The optimization on such high-dimensional state space is computationally intractable and hard to generalize on the test nodes. To enable the computation on common hardware, we propose to learn the joint distribution from cluster perspective. It is generally assumed nodes within the same cluster are highly connected while the edge connections between clusters are sparse. In other words, the node label distributions between clusters are close to be independent. We thereby

divide the graph into $M$ independent clusters $\{\mathcal{C}_1, \ldots, \mathcal{C}_M\}$ and factorize the joint distribution as:

$$p(y_1, \ldots, y_L \mid z_1, \ldots, z_L; \theta) := \prod_{m=1}^{M} p(\{y_i \mid v_i \in \mathcal{C}_m\} \mid \{z_i \mid v_i \in \mathcal{C}_m\}; \theta). \tag{3}$$

Although the i.i.d assumption on clusters reduces the computation complexity to some extent, the joint modeling on a subset of nodes is still impractical and is unfriendly to be adopted to infer the classes of test nodes. In this work, for each node representation-label pair $(z_i, y_i)$, we instead learn a joint conditional distribution $p(y_i, \bar{y}_m \mid z_i, \bar{z}_m; \theta)$, where $\bar{z}_m$ and $\bar{y}_m$ denote the statistical cluster feature and label, respectively. One of the simplest ways to construct the cluster feature and label is to average the node representations and labels from the training samples within the corresponding cluster, which is adopted in our method. The more advanced solution, like differentiable features and label vectors, could be used to learn the cluster statistics. Given the set of training nodes, MLE optimizes the joint-cluster conditional distribution as:

$$\hat{\theta}_{JC}\left(\{z_i, y_i\}_{i=1}^{L}\right) = \arg\max_{\theta} p(y_1, \ldots, y_L \mid z_1, \ldots, z_L; \theta) \tag{4a}$$

$$= \arg\max_{\theta} \prod_{m=1}^{M} p(\{y_i \mid v_i \in \mathcal{C}_m\} \mid \{z_i \mid v_i \in \mathcal{C}_m\}; \theta) \tag{4b}$$

$$= \arg\max_{\theta} \prod_{m=1}^{M} \prod_{i=1}^{|\mathcal{C}_m|} p(y_i, \bar{y}_m \mid z_i, \bar{z}_m; \theta) \tag{4c}$$

$$= \arg\max_{\theta} \sum_{m=1}^{M} \sum_{i=1}^{|\mathcal{C}_m|} \log p(y_i, \bar{y}_m \mid z_i, \bar{z}_m; \theta). \tag{4d}$$

Through the transition from Eq.(4b) to Eq.(4c), we decouple the distributions of connected nodes to facilitate computation but still keep the node-cluster relation to realize joint modeling. In this way, we can easily train on a set of individual nodes and extend the well-trained model to estimate the joint distribution of test samples. In this work, we use the graph clustering algorithm of METIS (Karypis & Kumar, 1998), which aims to construct the vertex partitions such that within clusters links are much more than between-cluster links to better capture the community structure of the graph. This partitioning manner is in line with our i.i.d assumption on clusters, where the between-cluster dependencies are negligible. Based on the above joint modeling, we introduce how to train models and infer node classes, and put the pseudo-code in Appendix A for further detailed information.

**Training with joint-cluster loss.** We design the joint-cluster cross-entropy loss to learn the node-cluster distribution. Let $f_\theta : (x_i, \bar{x}_m) \to (y_i, \bar{y}_m)$ be a model to map the node and cluster features into their corresponding joint label $y_i \bar{y}_m^\mathsf{T} \in \mathbb{R}^{c \times c}$. As shown in Figure 1, the model consists of an encoder (e.g., GNNs) to generate node representations and a classifier to predict the joint label. Recalling Section 2.1, the final representation of node $v_i$ is given by $z_i$. We adopt average pooling to define the cluster representation $\bar{z}_m = 1/L_m \sum_{k=1}^{L_m} z_i$ and the cluster label $\bar{y}_m = 1/L_m \sum_{k=1}^{L_m} y_i$, where $L_m$ is the number of labeled nodes within cluster $\mathcal{C}_m$. We then concatenate the node and cluster representations as the joint feature, which is fed into the classifier to predict joint label $y_i \bar{y}_m^\mathsf{T}$. Mathematically, the joint-cluster cross-entropy loss is defined as:

$$\mathcal{L}_{JC} = -\sum_{i=1}^{L} \{(y_i \bar{y}_m^\mathsf{T}) \cdot \log g_\phi(\mathrm{con}(z_i, \bar{z}_m)) + (\bar{y}_m y_i^\mathsf{T}) \cdot \log g_\phi(\mathrm{con}(\bar{z}_m, z_i))\}. \tag{5}$$

where $\mathrm{con}(\cdot, \cdot)$ is a vector concatenation operation ordered by the node embedding and its cluster embedding, $g_\phi$ is the classifier, and node $v_i$ belongs to cluster $\mathcal{C}_m$. The dot product and $\log$ function operate element-wisely. Notably, for the purpose of symmetric joint distribution modeling, at the second item of the above equation, we exchange the position of node and cluster embeddings to predict their label $\bar{y}_m y_i^\mathsf{T}$ (i.e., the transpose of $y_i \bar{y}_m^\mathsf{T}$).

**Node class inference in joint distribution.** Based on the joint distribution $p(y_i, \bar{y}_m \mid z_i, \bar{z}_m; \theta)$ between the node and its cluster, we aim to infer every individual node classes as in the standard

supervised learning framework. In other words, we have to recover the independent conditional distribution $p(y_i \mid z_i; \theta)$ and make a decision over the label state space $\mathbb{R}^c$. The direct solution is to marginalize the joint label along the cluster label dimension:

$$
\begin{aligned}
p(y_i \mid z_i; \theta) &= \int_{\mathbb{R}^d} \sum_{k=1}^{c} p\left(y_i, \bar{y}_m = k \mid z_i, \bar{z}; \theta\right) q\left(\bar{z}\right) d\bar{z} \\
&\approx \sum_{k=1}^{c} p\left(y_i, \bar{y}_m = k \mid z_i, \bar{z}_m; \theta\right).
\end{aligned}
\tag{6}
$$

$q\left(\bar{z}\right)$ denotes the continuous distribution of cluster representation. In practice, since the node is nearly independent to the other clusters, the approximation deduction in Eq.(6) only uses the dwelling cluster feature to obtain the marginalized distribution. As illustrated in Figure 1, given the two-dimensional prediction $p\left(y_i, \bar{y}_m \mid z_i, \bar{z}_m; \theta\right)$ corresponding to truth $y_i \bar{y}_m^{\mathsf{T}}$, we sum the prediction scores row wisely to estimate $p(y_i \mid z_i; \theta)$. Unlike the standard supervised learning, during model inference, we make use of the cluster reference signal to reason the node classes rationally and robustly. This merit is functionally similar to the in-context learning explored recently (Min et al., 2021), where a set of data-label pairs are concatenated with input to guide the language model to make more accurate decisions. As empirically studies in Appendix G, in graph data, we observe the joint distribution modeling provides better node classification accuracy compared with the simple concatenation.

## 4 RELATED WORK

A detailed discussion is provided in Appendix C. Two families of label dependency modeling are:

**Label propagation.** In the realm of GNNs, label propagation works on the assumption that nodes connected by an edge are likely to share the same label (Shi et al., 2020; Wang & Leskovec, 2020; Shi et al., 2020). It propagates and smooth node labels along with edge weights throughout the graph (Wang & Leskovec, 2021; Xie et al., 2022b), and then infer the unlabeled nodes effectively. *Difference compared to existing work*: While the label propagation often infer nodes without considering the node features (e.g., at post-processing phase), our joint-cluster loss could work with any GNN backbones to comprehensively learn the structure, feature, and label information end-to-end.

**Conditional random fields.** To leverage the label correlation in node classification, there has been previous art in combining conditional random fields (CRF) with GNNs. CGNF (Ma et al., 2018) learns the pairwise label correlation with pairwise energy function, a specific expression form of CRF, which is optimized to train GNNs. CRF-GNNs inserts CRF layer between the graph convolutional layers, which regularizes GNNs to preserve the label dependencies among nodes (Gao et al., 2019). *Difference compared to existing work*: ① While CRF-based methods focus on modeling the local label correlation of every linked node pair, we learn the global joint distribution of node and its cluster. ② Our proposals shows promising training scalability and inference efficiency. The CRF-based methods take the whole graph as input to propagate all the pairwise label correlations along edges. In contrast, we train and infer the joint distribution of target node only with one reference signal (i.e., cluster), which allows the batch training on large graphs (e.g., Amazon with millions of nodes).

## 5 EXPERIMENTS

In this section, we evaluate the proposed joint-cluster learning framework on 12 public datasets over 7 backbone models, to validate the effectiveness of generalization and robustness.

### 5.1 EVALUATION ON SMALL GRAPH DATASETS

**Implementation.** ▷ Datasets. We use the benchmark datasets Cora, CiteSeer, PubMed (Sen et al., 2008), DBLP (Bojchevski & Günnemann, 2017), and Facebook (Rozemberczki et al., 2021a) in the class-balanced setting, which is widely adopted to evaluate GNNs. Furthermore, we consider two more challenging node classification tasks. In particular, we conduct on LastFMAsia (Rozemberczki & Sarkar, 2020) and ogbn-arxiv (Hu et al., 2020) to evaluate the performance of our proposed joint-cluster loss on class-imbalanced environment; and we consider Chameleon, Squirrel, and Wisconsin to evaluate on heterophilic graphs (Rozemberczki et al., 2021b). ▷ Backbone models.

Table 1: Test Accuracy (%) for different models on class-balanced small datasets, where the best results are in bold. CE denotes the cross-entropy loss, and JC denotes our joint-cluster loss function.

| Model | Loss | Cora | CiteSeer | PubMed | DBLP | Facebook |
|-------|------|------|----------|--------|------|----------|
| GCN | CE | $81.70_{\pm 0.65}$ | $71.43_{\pm 0.47}$ | $79.06_{\pm 0.32}$ | $74.30_{\pm 1.94}$ | $73.91_{\pm 1.40}$ |
|     | JC | $\mathbf{83.51_{\pm 0.35}}$ | $\mathbf{72.97_{\pm 0.55}}$ | $\mathbf{79.80_{\pm 0.19}}$ | $\mathbf{75.10_{\pm 1.63}}$ | $\mathbf{74.64_{\pm 1.75}}$ |
| SGC | CE | $81.68_{\pm 0.52}$ | $71.85_{\pm 0.39}$ | $78.70_{\pm 0.38}$ | $74.30_{\pm 2.12}$ | $74.13_{\pm 2.13}$ |
|     | JC | $\mathbf{83.87_{\pm 0.79}}$ | $\mathbf{72.92_{\pm 0.16}}$ | $\mathbf{79.97_{\pm 0.25}}$ | $\mathbf{74.87_{\pm 1.81}}$ | $\mathbf{74.74_{\pm 1.96}}$ |
| SAGE | CE | $79.96_{\pm 0.44}$ | $69.94_{\pm 0.93}$ | $78.37_{\pm 0.72}$ | $70.59_{\pm 1.46}$ | $70.95_{\pm 2.26}$ |
|      | JC | $\mathbf{80.81_{\pm 0.63}}$ | $\mathbf{70.54_{\pm 1.49}}$ | $\mathbf{79.50_{\pm 1.02}}$ | $\mathbf{71.87_{\pm 2.07}}$ | $\mathbf{71.59_{\pm 1.78}}$ |
| GAT | CE | $83.22_{\pm 0.29}$ | $71.06_{\pm 0.40}$ | $78.54_{\pm 0.63}$ | $75.32_{\pm 2.62}$ | $76.34_{\pm 2.26}$ |
|     | JC | $\mathbf{83.77_{\pm 0.44}}$ | $\mathbf{71.61_{\pm 0.95}}$ | $\mathbf{79.35_{\pm 0.47}}$ | $\mathbf{76.92_{\pm 1.59}}$ | $\mathbf{77.46_{\pm 2.30}}$ |
| MLP | CE | $58.65_{\pm 0.97}$ | $60.41_{\pm 0.56}$ | $73.27_{\pm 0.35}$ | $47.95_{\pm 3.97}$ | $55.34_{\pm 2.60}$ |
|     | JC | $\mathbf{67.19_{\pm 0.62}}$ | $\mathbf{63.23_{\pm 0.87}}$ | $\mathbf{75.92_{\pm 0.39}}$ | $\mathbf{61.16_{\pm 3.63}}$ | $\mathbf{56.62_{\pm 2.42}}$ |

Table 2: Test F1(%) of different loss functions on the class-imbalanced datasets.

| Model | Loss | LastFMAsia | | | ogbn-arxiv | | |
|-------|------|----------|----------|-----------|----------|----------|-----------|
|       |      | F1-micro | F1-macro | F1-weight | F1-micro | F1-macro | F1-weight |
| GCN | CE | $84.91_{\pm 0.74}$ | $73.79_{\pm 1.28}$ | $84.60_{\pm 0.73}$ | $71.74_{\pm 0.29}$ | $51.80_{\pm 0.44}$ | $70.93_{\pm 0.33}$ |
|     | JC | $\mathbf{85.92_{\pm 0.41}}$ | $\mathbf{74.61_{\pm 1.02}}$ | $\mathbf{85.49_{\pm 0.43}}$ | $\mathbf{72.17_{\pm 0.24}}$ | $\mathbf{52.06_{\pm 0.15}}$ | $\mathbf{71.57_{\pm 0.18}}$ |
| SGC | CE | $84.82_{\pm 0.82}$ | $70.85_{\pm 1.36}$ | $84.22_{\pm 0.55}$ | $71.77_{\pm 0.14}$ | $50.75_{\pm 0.29}$ | $70.71_{\pm 0.21}$ |
|     | JC | $\mathbf{85.84_{\pm 0.45}}$ | $\mathbf{73.25_{\pm 1.17}}$ | $\mathbf{85.32_{\pm 0.40}}$ | $\mathbf{72.08_{\pm 0.15}}$ | $\mathbf{51.15_{\pm 0.26}}$ | $\mathbf{70.92_{\pm 0.15}}$ |
| MLP | CE | $68.91_{\pm 0.70}$ | $42.37_{\pm 1.45}$ | $67.06_{\pm 0.73}$ | $55.50_{\pm 0.23}$ | $33.93_{\pm 0.20}$ | $55.00_{\pm 0.19}$ |
|     | JC | $\mathbf{78.81_{\pm 0.69}}$ | $\mathbf{53.09_{\pm 1.71}}$ | $\mathbf{76.89_{\pm 0.69}}$ | $\mathbf{61.21_{\pm 0.16}}$ | $\mathbf{38.61_{\pm 0.23}}$ | $\mathbf{60.48_{\pm 0.11}}$ |

We take GCN (Kipf & Welling, 2017), SGC (Wu et al., 2019), GraphSage (Hamilton et al., 2017), GAT (Veličković et al., 2018) and MLP as base models to compare our proposals with the standard cross-entropy loss in the class-balanced setting. Due to space limit, we use GCN, SGC, and MLP to evaluate on the class-imbalanced and heterophilic environment. The details of datasets and backbone models are presented in Appendix B D. We run each experiment 10 times and report the mean values with standard deviation.

**Q: Whether our proposals outperform the standard supervised learning on the easy and small datasets?**    Yes, one key advantage of joint distribution modeling is to infer nodes more correctly with cluster references. We examine on class-balanced, class-imbalanced, and heterophilic datasets.

▷ Class-balanced graph datasets. The comparison results are collected in Table1, from which we make following observations. ❶ *The joint-cluster supervised learning exhibits significantly superior performances on all the backbone models.* Compared with the standard cross-entropy loss, our approach delivers the average improvements of 1.47%, 1.47%, 1.21%, 1.22% on models GCN, SGC, SAGE, and GAT, respectively. ❷ Interestingly, compared with the average improvement of 1.34% over GNN backbones, the more clear advantage of 10.5% is achieved in MLP architecture. That is because GNNs learn the single node class conditioned on aggregated features, while MLP decides the node label only based on its input feature. Moving a step forward, our proposals learn the comprehensive joint distribution of multiple node labels conditioned on their features aggregated from GNNs, which fully activates the model's generalization ability.

▷ Class-imbalanced graph datasets. As shown in Table2, we observe ❶ *the similar trend of performance enhancement in the imbalance setting.* We use imbalance ratio, $min_i\left(|\mathcal{T}_i|\right)/max_i\left(|\mathcal{T}_i|\right)$, to measure the extent of class imbalance, where $|\mathcal{T}_i|$ represents the number of nodes belonging to the $i$-th class. LastFMAsia and ogbn-arxiv are two extremely imbalanced datasets, whose imbalance rates are 1.0% and 0.1%, respectively. It is observed our joint-cluster learning framework obtains average improvements of 5.75% and 3.16% on LastFMAsia and ogbn-arxiv over the standard supervised learning. We attribute this result to the referential ability of the joint-cluster distribution modeling, which uses the cluster of neighbors when making decisions. The joint distribution weakens the over-confident prediction on the majority classes by assigning prediction confidence on other related minority classes, and thus ameliorates the generalization on them.

▷ Heterophilic graph datasets.    On the heterophilic graphs, the connected nodes tend to have the different classes and make the joint-distribution learning challenging via adding label noise.    Following the data split of Pei et al. (2020), we compare with vanilla cross-

entropy loss on three benchmark datasets. As shown in Table 3, ❶ *we observe our joint-cluster loss function consistently delivers great advantage with clear performance margin.* That is because the proposed joint-cluster distribution learning infer node label with the reference signal of whole cluster, instead of using the direct neighbors. This validates the effectiveness of adopting global cluster structure in joint distribution.

Table 3: Test accuracy (%) on heterophilic graphs.

| Model | Loss | Chameleon | Squirrel | Wisconsin |
|---|---|---|---|---|
| GCN | CE | $59.25_{\pm 2.81}$ | $48.93_{\pm 2.21}$ | $49.22_{\pm 3.77}$ |
| | JC | $\mathbf{68.87_{\pm 2.55}}$ | $\mathbf{56.76_{\pm 1.43}}$ | $\mathbf{50.39_{\pm 4.53}}$ |
| SGC | CE | $63.88_{\pm 2.78}$ | $53.79_{\pm 3.13}$ | $51.96_{\pm 4.23}$ |
| | JC | $\mathbf{71.91_{\pm 2.03}}$ | $\mathbf{61.99_{\pm 2.42}}$ | $\mathbf{52.23_{\pm 3.94}}$ |
| MLP | CE | $41.90_{\pm 1.51}$ | $29.23_{\pm 2.09}$ | $80.98_{\pm 5.12}$ |
| | JC | $\mathbf{50.09_{\pm 2.42}}$ | $\mathbf{32.37_{\pm 2.20}}$ | $\mathbf{81.96_{\pm 5.45}}$ |

## 5.2 EVALUATION ON LARGE GRAPH DATASETS

**Implementation.** ▷ Datasets. Two complex large datasets are adopted, i.e., Yelp and Amazon (Zeng et al., 2019), where each node contains multiple classes. ▷ Backbone models. We evaluate in two scalable sub-graph sampling models, i.e., GraphSAGE (Hamilton et al., 2017) and Cluster-GCN (Chiang et al., 2019), and in pre-computing-based model of SIGN (Frasca et al., 2020). The details of datasets and backbone models are provided in Appendix B D.

**Q: Whether our proposals can scale on the large datasets and boost model performance?** Yes, as reported in Table 4, ❶ *the joint-cluster learning framework generally obtains the best accuracy on the large-scale multi-class datasets.* Compared with the standard cross-entropy, our method obtains the average improvement of 0.75% and 0.39% on Yelp and Amazon, respectively. One exceptional cases is SIGN con-

Table 4: Test micro-F1(%) on large graph datasets.

| Model | Method | Yelp | Amazon |
|---|---|---|---|
| GraphSAGE | CE | $63.67_{\pm 0.38}$ | $75.65_{\pm 0.16}$ |
| | JC | $\mathbf{63.99_{\pm 0.46}}$ | $\mathbf{76.14_{\pm 0.29}}$ |
| Cluster-GCN | CE | $62.44_{\pm 0.52}$ | $76.12_{\pm 0.17}$ |
| | JC | $\mathbf{63.02_{\pm 0.68}}$ | $\mathbf{76.63_{\pm 0.27}}$ |
| SIGN | CE | $64.42_{\pm 0.07}$ | $\mathbf{80.22_{\pm 0.04}}$ |
| | JC | $\mathbf{64.95_{\pm 0.09}}$ | $80.09_{\pm 0.05}$ |

ducting on Amazon dataset. We speculate that one of the main reasons is the batch size, which is not large enough to obtain enough cluster statistics for the joint-cluster distribution modeling. The future work can use the trainable cluster feature and label to overcome this problem.

## 5.3 ROBUSTNESS UNDER ADVERSARIAL ATTACK

**Implementation.** Following the previous work, we use datasets including Cora, CiteSeer and PubMed to evaluate robustness under an untargeted adversarial graph attack. Specifically, we use the metattack (Sun et al., 2020) implemented in DeepRobust[1], a pytorch library, to generate attacked graphs by deliberately modifying the graph structure. The details are summarized in Appendix B.

**Q: Compared with vanilla training, whether the joint-distribution learning can ameliorate model's robustness under adversarial attack?** Yes, the comparison results are collected in Table 5, where we make the following observations to support our answers. ❶ *The joint-cluster learning framework achieves significant gain under all perturbation rates.* Compared with the independent decision making, our joint-cluster modeling takes the whole cluster as reference signals, which contains certain number of clean nodes to improve the robustness of class prediction. ❷ *The performance gain increases with the perturbation rates.* Specifically, the absolute improvements over the vanilla loss are 2.0%, 2.2%, 6.3%, 5.0% and 7.5% in the perturbation rates of 5%, 10%, 15%, 20%, and 25%. These results validate the effectiveness of cluster reference signal, which is structrually stable even under the acute attacks.

## 5.4 COMPARISON WITH LABEL DEPENDENCY MODELING RELATED WORK

**Q: Whether the joint-cluster supervised learning delivers the superior accuracy and efficiency compared with existing label dependency learning frameworks?** Yes, we examine it below.

▷ Comparison with CRF-based models. We consider two CRF-based models, i.e., CGNF and CRF-GCN, and collect the comparison results in Table 6. ❶ *Our proposals obtain the clear*

---

[1]https://github.com/DSE-MSU/DeepRobust

Table 5: Test accuracy (%) under metattack, where Ptb Rate means the perturbation percent.

| Datasets | Ptb Rate(%) | GCN | | SGC | | GAT | |
|---|---|---|---|---|---|---|---|
| | | CE | JC | CE | JC | CE | JC |
| Cora | 5% | $76.80_{\pm 0.87}$ | $\mathbf{78.84_{\pm 0.57}}$ | $76.28_{\pm 0.20}$ | $\mathbf{78.76_{\pm 0.45}}$ | $80.24_{\pm 0.54}$ | $\mathbf{80.76_{\pm 0.51}}$ |
| | 10% | $70.12_{\pm 1.42}$ | $\mathbf{74.65_{\pm 0.44}}$ | $69.29_{\pm 0.48}$ | $\mathbf{73.50_{\pm 0.57}}$ | $74.89_{\pm 1.46}$ | $\mathbf{75.49_{\pm 0.83}}$ |
| | 15% | $64.21_{\pm 1.92}$ | $\mathbf{72.24_{\pm 0.67}}$ | $65.05_{\pm 1.09}$ | $\mathbf{71.93_{\pm 0.51}}$ | $70.55_{\pm 1.19}$ | $\mathbf{71.63_{\pm 1.29}}$ |
| | 20% | $53.56_{\pm 1.98}$ | $\mathbf{59.77_{\pm 0.75}}$ | $57.14_{\pm 0.32}$ | $\mathbf{58.11_{\pm 0.83}}$ | $58.74_{\pm 1.60}$ | $\mathbf{59.45_{\pm 1.12}}$ |
| | 25% | $48.98_{\pm 1.58}$ | $\mathbf{53.89_{\pm 1.19}}$ | $51.18_{\pm 0.51}$ | $\mathbf{53.44_{\pm 1.04}}$ | $53.38_{\pm 1.14}$ | $\mathbf{55.46_{\pm 1.68}}$ |
| CiteSeer | 5% | $69.96_{\pm 0.82}$ | $\mathbf{70.15_{\pm 0.79}}$ | $71.87_{\pm 0.20}$ | $\mathbf{72.72_{\pm 0.62}}$ | $72.03_{\pm 1.08}$ | $\mathbf{73.96_{\pm 0.53}}$ |
| | 10% | $67.39_{\pm 0.74}$ | $\mathbf{68.51_{\pm 1.06}}$ | $68.19_{\pm 0.15}$ | $\mathbf{68.84_{\pm 0.50}}$ | $70.21_{\pm 0.82}$ | $\mathbf{71.10_{\pm 0.24}}$ |
| | 15% | $64.32_{\pm 0.93}$ | $\mathbf{67.23_{\pm 1.24}}$ | $65.01_{\pm 1.68}$ | $\mathbf{67.59_{\pm 0.79}}$ | $67.99_{\pm 1.43}$ | $\mathbf{70.39_{\pm 0.57}}$ |
| | 20% | $55.18_{\pm 1.67}$ | $\mathbf{57.59_{\pm 1.29}}$ | $56.38_{\pm 0.23}$ | $\mathbf{56.67_{\pm 0.88}}$ | $60.40_{\pm 1.41}$ | $\mathbf{61.57_{\pm 0.98}}$ |
| | 25% | $56.22_{\pm 2.27}$ | $\mathbf{61.54_{\pm 2.01}}$ | $55.94_{\pm 0.14}$ | $\mathbf{61.75_{\pm 0.92}}$ | $59.60_{\pm 2.18}$ | $\mathbf{60.74_{\pm 1.05}}$ |
| PubMed | 5% | $83.09_{\pm 0.10}$ | $\mathbf{83.17_{\pm 0.10}}$ | $78.12_{\pm 0.03}$ | $\mathbf{83.07_{\pm 0.07}}$ | $82.27_{\pm 0.19}$ | $\mathbf{82.97_{\pm 0.30}}$ |
| | 10% | $81.08_{\pm 0.18}$ | $\mathbf{81.27_{\pm 0.10}}$ | $71.16_{\pm 0.00}$ | $\mathbf{81.35_{\pm 0.06}}$ | $79.93_{\pm 0.16}$ | $\mathbf{81.81_{\pm 0.39}}$ |
| | 15% | $78.31_{\pm 0.28}$ | $\mathbf{78.71_{\pm 0.07}}$ | $67.16_{\pm 0.03}$ | $\mathbf{78.85_{\pm 0.06}}$ | $78.24_{\pm 0.13}$ | $\mathbf{80.08_{\pm 0.24}}$ |
| | 20% | $76.55_{\pm 0.34}$ | $\mathbf{76.90_{\pm 0.08}}$ | $63.88_{\pm 0.02}$ | $\mathbf{77.03_{\pm 0.09}}$ | $75.83_{\pm 0.27}$ | $\mathbf{78.02_{\pm 0.34}}$ |
| | 25% | $74.51_{\pm 0.50}$ | $\mathbf{75.05_{\pm 0.07}}$ | $61.10_{\pm 0.01}$ | $\mathbf{75.05_{\pm 0.11}}$ | $73.01_{\pm 0.35}$ | $\mathbf{75.61_{\pm 0.32}}$ |

Table 6: Accuracy (%), training time (s), and inference time (s) comparisons with CRF-based models. Since CRF-GCN does not provide code, the accuracy is directly reported and the time is omitted.

| Methods | Cora | | | CiteSeer | | | PubMed | | |
|---|---|---|---|---|---|---|---|---|---|
| | Accuracy | Training | Inference | Accuracy | Training | Inference | Accuracy | Training | Inference |
| GCN | 81.70 | 0.002 | 0.001 | 71.43 | 0.002 | 0.001 | 79.06 | 0.008 | 0.001 |
| CGNF | 83.2 | 0.389 | 0.181 | 72.2 | 0.240 | 0.093 | 79.4 | 7.523 | 2.959 |
| CRF-GCN | 82.8 | – | – | 72.1 | – | – | 79.2 | – | – |
| GCN+JC | **83.51** | 0.004 | 0.001 | **72.97** | 0.005 | 0.001 | **79.80** | 0.018 | 0.001 |

*performance gains even compared with SOTA models encoding label dependency.* Particularly, the absolute improvements are 0.4%, 1.1% and 0.5% on Cora, Citeseer, and Pubmed, respectively. These baselines predict the target node by accounting the label dependencies from all the connected neighbors. In contrast, we only take the cluster as reference signal to learn the joint distribution, which is simple but shows great generalization. ❷ *Our proposals consume much less training and inference times, which are comparable to vanilla GCN.* While we only consider the cluster in joint distribution, the CRF-based models learn the target node together with all its neighbors burdensomely.

▷ Concatenating with label propagation. C&S (Huang et al., 2020) is proposed to smooth node labels at the post-processing phase of MLP model. Prior to such post-processing, our joint distribution labeling can be plugged in to better prepare MLP by learning the label correlations of nodes. We examine our thoughts in Table7. ❶ *It is observed that over all the larger datasets (i.e., except Cora and Citeseer), MLP can evidently benefit from the joint-cluster loss.* On the small datasets, the stacking of C&S and joint loss will make the node labels overly similar over the whole graph and degrade model performance.

Table 7: Performance of C&S with the MLP trained by cross-entropy loss and joint-cluster loss.

| Methods | Cora | CiteSeer | PubMed | DBLP | Facebook | LastFMAsia | Arxiv |
|---|---|---|---|---|---|---|---|
| MLP+CE | $58.65_{\pm 0.97}$ | $60.41_{\pm 0.56}$ | $73.27_{\pm 0.35}$ | $47.95_{\pm 3.97}$ | $55.34_{\pm 2.60}$ | $68.91_{\pm 0.70}$ | $55.50_{\pm 0.23}$ |
| MLP+JC | $67.19_{\pm 0.62}$ | $63.23_{\pm 0.87}$ | $75.92_{\pm 0.39}$ | $61.16_{\pm 3.63}$ | $56.62_{\pm 2.42}$ | $78.81_{\pm 0.69}$ | $63.13_{\pm 0.10}$ |
| MLP+CE+C&S | $\mathbf{80.05_{\pm 0.46}}$ | $\mathbf{70.36_{\pm 0.44}}$ | $77.08_{\pm 0.26}$ | $71.19_{\pm 2.59}$ | $67.48_{\pm 4.60}$ | $85.73_{\pm 0.61}$ | $68.58_{\pm 0.05}$ |
| MLP+JC+C&S | $77.37_{\pm 0.65}$ | $69.01_{\pm 0.93}$ | $\mathbf{77.91_{\pm 0.43}}$ | $\mathbf{73.23_{\pm 0.86}}$ | $\mathbf{69.39_{\pm 3.53}}$ | $\mathbf{87.26_{\pm 0.54}}$ | $\mathbf{70.06_{\pm 0.09}}$ |

## 5.5 IN-DEPTH DISCUSSION OF JOINT-CLUSTER SUPERVISED LEARNING

**Q: How the joint-cluster distribution modeling learns to concentrate node embeddings of the same class (cluster) within compact space?** We visualize the node representations learned by cross-entropy loss and joint-cluster loss in Figure 2. Different from the vanilla loss, our joint-cluster loss exhibits 2D projections with more coherent shapes of clusters. One of the possible reasons is the node representations are learned to embrace their corresponding clusters in the joint modeling.

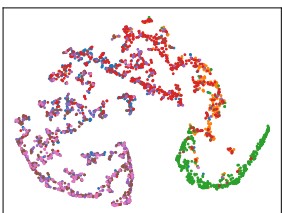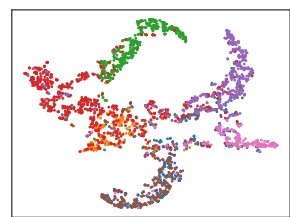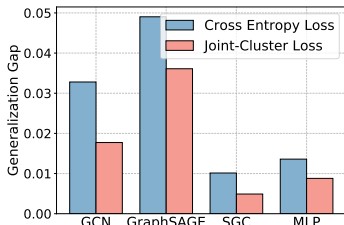

Figure 2: Left, Middle: Node representation visualization by t-SNE (Van der Maaten & Hinton, 2008) for 8-layer GCN trained by cross-entropy loss (left) and joint-cluster loss (middle) on Cora. Right: Normalized comparison of the gap between train and test losses on ogbn-arxiv.

**Q: Whether the joint learning avoids the overfitting on training set.** The model's generalization ability is commonly measured by the gap between training loss and test loss. The smaller the gap is, the better the model can be free from the overfitting but generalizes on the testing set. We plot such a loss gap in the right part of Figure 2, where joint-cluster loss generally has a smaller gap.

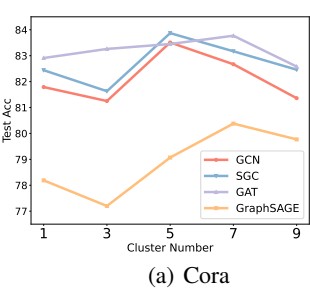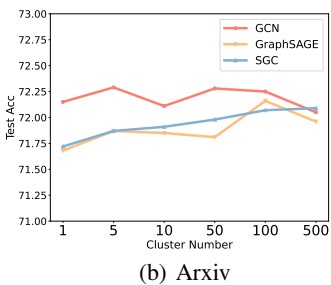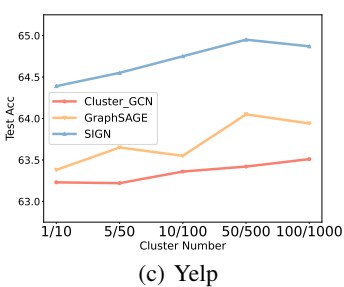

| (a) Cora | (b) Arxiv | (c) Yelp |
|---|---|---|

Figure 3: Hyperparameter effect of the cluster number in the joint-cluster supervised learning. Note that a/b in Yelp, a denotes cluster number in Cluster_GCN and GraphSAGE, and b represents cluster number in SIGN, which uses a larger batch size.

**Q: How is the sensitivity of the joint-cluster learning framework to the cluster number?** Fig.3 shows the hyperparameter effect of the cluster number on both small and large datasets. We observe the joint-cluster loss benefits from a suitable number in a smaller dataset Cora. Yet, we notice the performances are stable as the cluster number changes in larger datasets, such as Arxiv and Yelp.

**Q: Does our joint-cluster learning framework require expensive memory cost compared to standard supervised learning framework?** We examine this question in Table 8. It is found that our framework requires little cost on most models except GAT, which brings the non-negligible improvements in node classification accuracy and robustness over adversarial attack. Although GAT requires a higher cost due to its complex attention mechanism, this is still acceptable compared with the benefits.

Table 8: Occupied memory (ratio) of JC loss compared with vanilla loss.

| Model | Cora | CiteSeer | PubMed |
|---|---|---|---|
| GCN | $1.01\times$ | $1.04\times$ | $1.00\times$ |
| SGC | $1.06\times$ | $1.05\times$ | $1.00\times$ |
| MLP | $1.01\times$ | $1.05\times$ | $1.02\times$ |
| SAGE | $1.03\times$ | $1.08\times$ | $1.07\times$ |
| GAT | $1.70\times$ | $1.39\times$ | $1.56\times$ |

## 6 CONCLUSION

In this paper, we hypothesize that the independent conditional distribution of node labels is not in line with the graph-structured data, where nodes tend to connect with "similar" neighbors and linked nodes have complicated relationships. Based on the i.i.d assumption, the supervised learning with standard cross-entropy loss fails to fully activate the model's ability in generalizing over a test set as well as defending adversarial attacks. Motivated by the label dependencies between nodes and their corresponding clusters, we have presented the joint-cluster supervised learning framework for the training and inference in graph data. This new paradigm learns the joint distribution of nodes and their cluster labels conditioned on their features, and introduces the joint-cluster cross-entropy loss. The extensive experiments demonstrate that our model can boost the node classification performance of GNN models and simple MLP architecture compared to the standard supervised learning on a wide range of real-world datasets. The limitations and interesting future work are discussed in Appendix H.

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

## A  ALGORITHM

---

**Algorithm 1:** Joint-Cluster Learning Framework

---

**Input:** Adjacent matrix $\mathbf{A}$, features matrix $\mathbf{X}$, the set of labeled nodes $\mathbf{V}_L$ and their labels $\mathbf{Y}_L$, encoder $f$, classifier $g$

**Output:** Predicted labels of unlabeled nodes

1 Partition graph nodes into $M$ clusters $\mathbf{C}_1, \mathbf{C}_2, ..., \mathbf{C}_M$ by METIS;

2 **for** *each cluster* **do**

3      $\bar{\mathbf{Y}}_m = 1/L_m \sum_{k=1}^{L_m} \mathbf{Y}_k$;  $/ *$ Calculate cluster_label according to $\mathbf{Y}_L. * /$

4 **end**

5 **for** $i = 1; i \leq$ *max iteration epoch*; $i{+}{+}$ **do**

6      $\mathbf{Z} = f_\theta (\mathbf{A}, \mathbf{X})$;  $/ *$ Update node embedding.*/

7      $\bar{\mathbf{Z}}_m = 1/L_m \sum_{k=1}^{L_m} \mathbf{Z}_k$;  $/ *$ Update cluster_embeddings. $* /$

8      Update joint_embeddings $\mathbf{Z}_{jc}$ according to $\mathbf{Z}_L$ and $\bar{\mathbf{Z}}_m$

9      $\hat{\mathbf{Y}}_{jc} = g_\phi (\mathbf{Z}_{jc})$;  $/ *$ Obtain joint distribution prediction of Node and Cluster. $* /$

10      $\mathbf{Y}_{jc} = \mathbf{Y}_i \bar{\mathbf{Y}}_m^\top$;  $/ *$ Joint label of node and cluster. $* /$

11      Calculate joint-cluster loss. $\mathbf{L}_{jc}$

12      $\bigtriangledown_{\theta,\phi} [\mathcal{L}_{jc}]$

13 **end**

---

## B  THE STATISTICS OF DATASETS

Table 9 contains the statistics for the nine datasets used in our experiments for node classification. Experiments are under the single-class and multi-class setting. For single-class classification task, we conduct the experiments on an online social network (LastFMAsia (Rozemberczki & Sarkar, 2020)), a webpage dataset(Wisconsin[2]), three page-page networks (Facebook (Rozemberczki et al., 2021a), Chameleon and Squirrel (Rozemberczki et al., 2021b)) and citation networks, including Cora, CiteSeer, PubMed (Kipf & Welling, 2017), DBLP (Bojchevski & Günnemann, 2017) and ogbn-arxiv (Hu et al., 2020). For multi-class classification task, we use businesses types network based on customer reviewers and friendship (Yelp (Zeng et al., 2019)), and product network based on buyer reviewers and interactions (Amazon (Zeng et al., 2019)). Furthermore, the statistics of the datasets used in adversarial attack in Table 10.

Next, we will introduce in detail the data split. We follow the standard split proposed by  (Kipf & Welling, 2017) on three citation networks, including Cora, CiteSeer, and PubMed. For DBLP and Facebook, we use 20 labeled nodes per class as the training set, 30 nodes per class for validation, and the rest for testing. In addition, we conduct the experiments on LastFMAsia and ogbn-arxiv to further evaluate the performance of our proposed joint-cluster loss on imbalanced datasets. For LastFMAsia, we randomly split 25%/25%/50% of nodes for training, validation, and testing. For ogbn-arxiv, we follow the standard split proposed by Hu et al. (2020). For heterophilic graph datasets(Chameleon, Squirrel and Wisconsin), we fellow the data split of Pei et al. (2020). For two large multi-class datasets proposed by Zeng et al. (2019), including Yelp and Amazon, whose node numbers are 716K and 1598K. Following Zeng et al. (2019), we use the same data split to stay our focus on the design of the objective function and conduct a fair comparison with independent cross-entropy loss. For robustness experiments, following previous works (Jin et al., 2020b), we only consider the largest

---

[2]http://www.cs.cmu.edu/afs/cs.cmu.edu/project/theo-11/www/wwkb

Table 9: Statistics of datasets used in experiments ("m" stands for multi-class classification, and "s" for single-class).

| Datasets | Nodes | Edges | Features | Classes |
|----------|-------|-------|----------|---------|
| Cora | 2,708 | 5,429 | 1,433 | 7(s) |
| CiteSeer | 3,327 | 4,732 | 3,703 | 6(s) |
| PubMed | 19,717 | 44,338 | 500 | 3(s) |
| DBLP | 17,716 | 105,734 | 1,639 | 4(s) |
| Facebook | 22,470 | 342,004 | 128 | 4(s) |
| LastFMAsia | 7,624 | 55,612 | 128 | 18(s) |
| ogbn-arxiv | 169,343 | 1,166,243 | 128 | 40(s) |
| Chameleon | 2,277 | 36,101 | 2,325 | 5(s) |
| Squirrel | 5,201 | 217,073 | 2,089 | 5(s) |
| Wisconsin | 251 | 499 | 1,703 | 5(s) |
| Yelp | 716,847 | 6,977,410 | 300 | 100(m) |
| Amazon | 1,598,960 | 132,169,734 | 200 | 107(m) |

Table 10: Following Jin et al. (2020b), we only consider the largest connected component (LCC).

| Datasets | Nodes | Edges | Features | Classes |
|----------|-------|-------|----------|---------|
| Cora | 2,485 | 5,069 | 1,433 | 7 |
| CiteSeer | 2,110 | 3,668 | 3,703 | 6 |
| PubMed | 19,717 | 44,338 | 500 | 3 |
| Polblogs | 1,222 | 16,714 | / | 2 |

connected component (LCC) in the adversarial graphs, and randomly split 10%/10%/80% of nodes for training, validation, and testing.

## C  OTHER RELATED WORK

**Graph neural networks.**  Existing GNNs follow the neighborhood aggregation strategy, which iteratively updates the node representation by aggregating the representations of its neighboring nodes and combining them with its representations (Xu et al., 2018). Numerous variants of GNNs have been proposed to achieve outstanding performances in a wide variety of graph-based tasks, such as graph clustering (Bo et al., 2020; Liu et al., 2022b), node classification (Bruna et al., 2013; Kipf & Welling, 2017) and graph classification (Zhou et al., 2021; Lee et al., 2019). In order to deal with large-scale graph datasets, researchers have proposed some scalable graph learning methods (Chiang et al., 2019; Duan et al., 2022).

**Graph adversarial attack.**  Graph adversarial attack refers to the process of manipulating or perturbing the nodes, edges, or features in a graph to deceive or mislead graph-based learning models(Chen et al., 2020; Jin et al., 2020a). These attacks can be categorized into different types, such as structural attacks that modify the graph topology (Xu et al., 2019; Wang & Gong, 2019; Li et al., 2020), feature-based attacks that manipulate node features (Xue et al., 2021; Liu et al., 2022a), and hybrid attacks that combine both (Zhang et al., 2022; Xie et al., 2022a; Ma et al., 2022). Compared with cross-entropy loss, our joint-cluster loss can refer to similar nodes in the process of loss optimization and inference, which can effectively alleviate the impact of graph attacks.

## D  DESCRIPTION OF BACKBONE MODELS

We evaluate our joint-cluster supervised learning framework on differnet GNN models and scalable graph learning backbones:

- **GCN (Kipf & Welling, 2017):** GCN is a convolutional neural network which utilizes the structural information of graphs by message passing mechanism.

- **SGC (Wu et al., 2019):** SGC eliminates the nonlinearities of GCN and collapses the weight matrix into a weight matrix.

- **MLP:** MLP is a simple neural network that maps a set of input vectors to a set of output vectors.

- **GAT (Veličković et al., 2018):** GAT learns edge weights in graph domain through the attention mechanism and achieves significant performance.

- **GraphSAGE (Hamilton et al., 2017):** Graphsage obtains neighbor nodes through sampling strategies and expresses node representation through neighbor aggregation operations.

- **Cluster_GCN (Chiang et al., 2019):** Cluster_GCN is a fast and efficient mini-batch training algorithm that preserve structural information within a batch by exploiting the graph clustering structure.

- **SIGN (Frasca et al., 2020):** SIGN is amenable to efficient precomputation by using graph convolutional filters of different size, achieving fast training and inference.

## E  IMPLEMENTATION

Following the experimental settings of original papers, for GAT[3], we choose the model parameters by utilizing an early stopping strategy with a patience of 100 epochs on classification loss. For other GNN models[4,5], we utilize the model parameters which perform best on the validation set for testing. The remaining hyper-parameters including learning rate, dropout and weight decay are tuned for different models. Scalable graph learning methods are executed based on the official examples of PyTorch Geometric[6,7]. We further implement joint-cluster loss over each backbone framework. Because Graphsage and SIGN divide the batch, it is impossible to guarantee that the nodes in the same batch are adjacent. Therefore, in order to ensure fairness, for the joint-cluster loss of the large-scale graph learning methods, we use the manner of randomly assigning clusters to the nodes.

## F  MULTI-CLASS TASK DESIGN

We introduced the framework design of single-class classification task in the paper. In short, for the single-class setting, joint-cluster learning framework expands a $c$-class classification task into a $c^2$-class classification task. The multi-class setting is slightly different from single-class. The number of clsses $c$ in the multi-classification task represents $c$ binary classification tasks. We extend each two-class classification task to a four-class classification task for nodes and clusters, and use cross-entropy loss to optimize each four-class classification. So the output dimension of the classifier is $4c$.

## G  ADDITIONAL EXPERIMENTS

**In-context learning.**  We conduct experiments to demonstrate the effect of joint distribution modeling in joint-cluster supervised learning framework. For each experiment, we compare the model trained by standard supervised learning, in-context strategy and joint-cluster learning framework. For in-context learning, we use the same input as the joint-cluster framework, the output is a c-dimensional vector, and the node label is used as the ground truth. As shown in Table 11, we observe that in-context strategy does not get a stable accuracy improvement. We guess that in-context strategy requires the cluster label should be sharp and the node label should be consistent with the cluster label, which will cause the model to be limited by the division of clusters. Our joint-cluster framework learns the joint distribution of nodes and clusters, which will learn potentially complex relationships between nodes, not just similarities.

---

[3]https://github.com/pyg-team/pytorch_geometric/blob/master/examples/gat.py

[4]https://github.com/tkipf/pygcn

[5]https://github.com/Tiiiger/SGC

[6]https://github.com/pyg-team/pytorch_geometric/blob/master/examples/cluster_gcn_ppi.py

[7]https://github.com/pyg-team/pytorch_geometric/blob/master/examples/sign.py

Table 11: Test Accuracy (%) for different models on five datasets. In addition, we show the best results in bold. We run 10 times and report the mean ± standard deviation. CE denotes the standard cross-entropy loss, IC denotes in-context learning strategy, and JC denotes our joint-cluster learning framework.

| Model | Loss | Cora | CiteSeer | PubMed | DBLP | Facebook |
|-------|------|------|----------|--------|------|----------|
| GCN | CE | $81.70_{\pm0.65}$ | $71.43_{\pm0.47}$ | $79.06_{\pm0.32}$ | $74.30_{\pm1.94}$ | $73.91_{\pm1.40}$ |
|  | IC | $81.56_{\pm0.25}$ | $70.08_{\pm0.56}$ | $79.37_{\pm0.46}$ | $72.53_{\pm2.55}$ | $70.35_{\pm1.86}$ |
|  | JC | $\mathbf{83.51_{\pm0.35}}$ | $\mathbf{72.97_{\pm0.55}}$ | $\mathbf{79.80_{\pm0.19}}$ | $\mathbf{75.10_{\pm1.63}}$ | $\mathbf{74.64_{\pm1.75}}$ |
| SGC | CE | $81.68_{\pm0.52}$ | $71.85_{\pm0.39}$ | $78.70_{\pm0.38}$ | $74.30_{\pm2.12}$ | $74.13_{\pm2.13}$ |
|  | IC | $81.87_{\pm0.51}$ | $69.41_{\pm0.79}$ | $79.20_{\pm0.43}$ | $71.40_{\pm1.07}$ | $68.15_{\pm3.30}$ |
|  | JC | $\mathbf{83.87_{\pm0.79}}$ | $\mathbf{72.92_{\pm0.16}}$ | $\mathbf{79.97_{\pm0.25}}$ | $\mathbf{74.87_{\pm1.81}}$ | $\mathbf{74.74_{\pm1.96}}$ |
| SAGE | CE | $79.96_{\pm0.44}$ | $69.94_{\pm0.93}$ | $78.37_{\pm0.72}$ | $70.59_{\pm1.46}$ | $70.95_{\pm2.26}$ |
|  | IC | $78.70_{\pm1.13}$ | $67.52_{\pm0.96}$ | $78.50_{\pm0.58}$ | $70.17_{\pm3.29}$ | $69.75_{\pm1.75}$ |
|  | JC | $\mathbf{80.81_{\pm0.63}}$ | $\mathbf{70.54_{\pm1.49}}$ | $\mathbf{79.50_{\pm1.02}}$ | $\mathbf{71.87_{\pm2.07}}$ | $\mathbf{71.59_{\pm1.78}}$ |
| GAT | CE | $83.22_{\pm0.29}$ | $71.06_{\pm0.40}$ | $78.54_{\pm0.63}$ | $75.32_{\pm2.62}$ | $76.34_{\pm2.26}$ |
|  | IC | $83.21_{\pm0.32}$ | $\mathbf{71.43_{\pm0.47}}$ | $78.38_{\pm0.22}$ | $74.10_{\pm1.59}$ | $72.49_{\pm2.34}$ |
|  | JC | $\mathbf{83.77_{\pm0.44}}$ | $70.18_{\pm0.86}$ | $\mathbf{79.35_{\pm0.47}}$ | $\mathbf{76.92_{\pm1.59}}$ | $\mathbf{77.46_{\pm2.30}}$ |
| MLP | CE | $58.65_{\pm0.97}$ | $60.41_{\pm0.56}$ | $73.27_{\pm0.35}$ | $47.95_{\pm3.97}$ | $55.34_{\pm2.60}$ |
|  | IC | $64.27_{\pm0.43}$ | $62.27_{\pm1.69}$ | $75.74_{\pm0.46}$ | $58.83_{\pm2.31}$ | $56.53_{\pm3.20}$ |
|  | JC | $\mathbf{67.19_{\pm0.62}}$ | $\mathbf{63.23_{\pm0.87}}$ | $\mathbf{75.92_{\pm0.39}}$ | $\mathbf{61.16_{\pm3.63}}$ | $\mathbf{56.62_{\pm2.42}}$ |

Table 12: Node classification accuracy (%) under metattack.

| Datasets | Ptb Rate(%) | GCN | | SGC | | GAT | |
|----------|-------------|-----|-----|-----|-----|-----|-----|
|  |  | CE | JC | CE | JC | CE | JC |
| Polblogs | 5% | $72.70_{\pm0.60}$ | $\mathbf{74.15_{\pm0.39}}$ | $74.44_{\pm0.38}$ | $\mathbf{76.64_{\pm0.51}}$ | $76.56_{\pm0.74}$ | $\mathbf{78.65_{\pm0.88}}$ |
|  | 10% | $71.90_{\pm0.69}$ | $\mathbf{78.37_{\pm3.57}}$ | $70.46_{\pm0.29}$ | $\mathbf{78.70_{\pm3.81}}$ | $72.42_{\pm0.69}$ | $\mathbf{76.79_{\pm1.11}}$ |
|  | 15% | $67.92_{\pm0.78}$ | $\mathbf{70.53_{\pm0.53}}$ | $55.99_{\pm1.85}$ | $\mathbf{72.64_{\pm1.24}}$ | $61.13_{\pm5.36}$ | $\mathbf{69.04_{\pm2.97}}$ |
|  | 20% | $57.76_{\pm0.37}$ | $\mathbf{62.84_{\pm0.93}}$ | $51.94_{\pm0.11}$ | $\mathbf{65.60_{\pm1.77}}$ | $51.96_{\pm0.17}$ | $\mathbf{52.04_{\pm0.16}}$ |
|  | 25% | $56.17_{\pm2.11}$ | $\mathbf{64.87_{\pm0.96}}$ | $52.02_{\pm0.46}$ | $\mathbf{61.67_{\pm4.12}}$ | $49.46_{\pm2.41}$ | $\mathbf{52.04_{\pm1.15}}$ |

**Robustness under adversarial attack.** To verify the robustness of our framework, we use one blog graph (Polblogs (Jin et al., 2020b)) commonly used in previous studies, whose node features are not available. we set the attribute matrix to $N * N$ identity matrix. The statistics of Polblogs is shown in Table 10. We randomly split 10%/10%/80% of nodes for training, validation, and testing. As shown in Table 12, our joint-cluster loss usually outperforms cross-entropy loss under different perturbation rates. For instance, our method improves SGC over 29% at 15% perturbation rate. Complementary experiments further demonstrate the advantages of our proposed framework in terms of robustness.

**Over-smoothing.** Our framework can alleviate over-smoothing. As shown in Figure 2 of manuscript, in a 8-layer GCN, our framework can exhibit 2D projection of node embeddings with more coherent shapes of clusters. In addition, we leverage GCN as the backbone networks, and compare joint-cluster loss with cross-entropy loss by considering the layer numbers of 2, 4, 8, 16, and 32. As shown in the Figure 4, our approach almost delivers the better node classification accuracies. That is because our framework separates the node distribution modeling of different clusters, which could relieve the over-smoothing issue to some extent.

**Efficiency analysis.** We use METIS to efficiently perform cluster division at pre-processing stage for small graphs with thousands of nodes, which takes less than five seconds. For the batch training on large graphs, we use random clustering on sampled training nodes and do not require clustering time cost. Next we show the training time and training memory per epoch for vanilla cross-entropy (CE) loss and our joint-cluster (JC) loss in Table 13. It is found that computational time overhead and memory cost are extremely marginal, which brings the non-negligible improvements in node classification accuracy and robustness over adversarial attack.

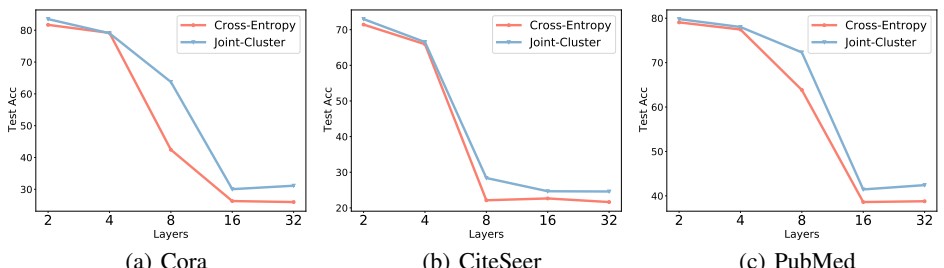

(a) Cora        (b) CiteSeer        (c) PubMed

Figure 4: Over-smoothing analysis about the model depth for node classification.

Table 13: The efficiency analysis of the training time and training memory.

| Datasets | Methods | Cora | | CiteSeer | | PubMed | |
|---|---|---|---|---|---|---|---|
| | | Time(s) | Memory(MB) | Time(s) | Memory(MB) | Time(s) | Memory(MB) |
| GCN | CE | 0.002 | 88.29 | 0.002 | 184.90 | 0.016 | 3061.18 |
| | JC | 0.004 | 89.16 | 0.005 | 191.53 | 0.018 | 3065.76 |
| SGC | CE | 0.002 | 60.55 | 0.002 | 142.67 | 0.002 | 1577.18 |
| | JC | 0.003 | 64.39 | 0.005 | 149.24 | 0.007 | 1581.75 |
| MLP | CE | 0.001 | 60.55 | 0.002 | 142.67 | 0.002 | 1577.18 |
| | JC | 0.004 | 61.15 | 0.004 | 149.24 | 0.006 | 1603.10 |
| SAGE | CE | 0.005 | 49.54 | 0.005 | 148.00 | 0.006 | 147.51 |
| | JC | 0.008 | 51.16 | 0.008 | 159.22 | 0.008 | 157.84 |
| GAT | CE | 0.005 | 61.60 | 0.005 | 157.43 | 0.006 | 246.93 |
| | JC | 0.007 | 104.99 | 0.008 | 218.44 | 0.011 | 385.76 |

## H    LIMITATIONS AND FUTURE WORK

Although our framework achieves promising experimental justifications, it suffers from the computation inefficiency issue. Compared with the standard supervised learning, the joint-cluster distribution modeling expands a $c$-classes node classification task into a $c^2$-classes prediction problem. Consequently, we require the larger memory and more expensive time cost especially for the graph data with a large number of node classes. However, this computation challenge can be relieved by reformulating the $c^2$-classes prediction problem to a $2c$-classes setting, where the ground-truth probability values are described by the corresponding node or cluster labels.

In the future work, we will explore the joint-cluster supervised learning on a broad range of potential applications, such as graph classification or link prediction. In addition, the correlation between samples is the biggest challenge in modeling real problems using probability theory, especially in graph data. We expect more studies and exploration on more intermediate factorizations between i.i.d and fully joint learning about the graph domain. We believe that the joint distribution learning will continue to be a promising research area.

