# OpenReview forum: "Rethinking Independent Cross-Entropy Loss For Graph-Structured Data"
_ICLR.cc/2024/Conference — Submitted to ICLR 2024_

### Official Review · Reviewer_8QYJ · 2023-10-30

**Soundness:** 3 good
**Presentation:** 3 good
**Contribution:** 2 fair
**Rating:** 5
**Confidence:** 4

**Summary:**

The paper proposes joint-cluster supervised learning for graph neural networks. Instead of adopting the cross-entropy loss for each node independently, the paper models the joint distribution of node and cluster labels, given their respective representations. Extensive experiments are conducted across multiple benchmarks, demonstrating the proposed loss can boost the performance of different backbone GNNs and robustness against adversarial attack.

**Strengths:**

1. The proposed joint modeling of node and cluster is novel and sound.

2. The scope of the experimental evaluation is broad, including small graphs and large graphs.

3. The empirical analyses are comprehensive in terms of necessary discussions, comparisons, and visualizations.

**Weaknesses:**

1. The backbones adopted for experiments are mostly not those that perform the best on these benchmarks. It would be more convincing to see how the proposed loss boost the performance of strong GNN models, e.g., GCNII on Cora, that may give rise to sota performance.

2. Similar concern to 1 also exists for analyses like Table 6 (with GCN) and Table 7 (with MLP).


[1] Chen et al. Simple and deep graph convolutional networks. In ICML.

**Questions:**

1. Could more results with stronger backbones be provided on these datasets? e.g., GCNII on Cora.

2. It is vague how this technique helps improve the best models that prevail on different tasks. For example, at least a comprehensive table which enumerates most recent or best performing methods on several datasets should be presented to give the readers an overview how this approach situate in the rich literatures in GNN-based node classification.

---

### Official Review · Reviewer_npyM · 2023-11-01

**Soundness:** 2 fair
**Presentation:** 3 good
**Contribution:** 3 good
**Rating:** 6
**Confidence:** 3

**Summary:**

The paper studies the problem of discrepancy between the non-i.i.d. property of GNN and the MLE learning. It proposed a new loss function to address the problem and its performance is demonstrated by extensive experiments.

**Strengths:**

1. The studied problem is important.
2. The idea of the proposed method is novel.
3. The method is effective in comparison to the baseline cross-entropy loss.

**Weaknesses:**

1. Some notations or definitions haven't been clearly explained. See the questions.
2. The discussion about the connection between (5) and (4d) is missing. This makes it difficult to follow (5).

**Questions:**

1. More explanation about the equivalance between (4c) and (4d) should be provided.
2. In the definitions of $\bar{z}_m$ and $\bar{y}_m$, there are two indices $k$ and $i$ that are confusing.
3. In (5), $y _i\bar{y} _m^\top$ is a matrix, which is not consistent with the shape of output of $g _\phi$.
4. What are the labeling rates for the datasets in Table 1? How does labeling rate influence the classification accuracy?
5. How did the authors determine the hyperparameters of the compared methods?
6. It is not clear why the improvement on balanced data is higher than imbalanced data.

---

> ### Author Response · Authors · 2023-11-20
> **Response to Reviewer npyM**
>
> We really appreciate the reviewer for the agreements on our technical contributions and presentation. We would like to address the concerns one by one in the following.
>
> **W1 and W2:** Thanks for your constructional suggestions. We will revise the paper according to the questions and clarify the connection between (5) and (4d) to facilitate readers' understanding of the relationship between maximum likelihood estimation and cross-entropy loss.
>
> **Q1:** We will introduce the transition from (4c) to (4d). We aim to learn the parameters $\theta$ to maximize the likelihood of the observed conditional distribution $p\left(y_{1}, \ldots, y_{L} \mid z_{1}, \ldots, z_{L} ; \theta\right)$, known as maximum likelihood estimation. To simplify the computation of the likelihood function, it is common practice to use the logarithm of the likelihood function, which transforms the product of probabilities into a sum (taking the logarithm does not alter the optimal solution of the likelihood).
>
> **Q2:** Thanks for your pointing out. We will fix this mistask and use $i$ as the indice uniformly.
>
> **Q3:** We flatten the matrix $y_i y_m^{\top} \in \mathbb{R}^{C\times C}$ into shape ${1\times C^2}$, and the shape of output of $g_\phi$ is ${n\times C^2}$, where $n$ represents the number of nodes and $c$ represents the number of classes.
>
> **Q4:** In Table 1, for citation networks(Cora, CiteSeer and PubMed), following the widely used standard split proposed by [1], we use 20 labeled nodes per class for training, 500 nodes for validation and 1000 nodes for testing. For DBLP and Facebook, we use 20 labeled nodes per class as the training set, 30 nodes per class for validation, and the rest for testing. To verify the the influence of labeling rate, we compare with vanilla cross-entropy (CE) loss on three citation networks. We randomly select five and ten labeled nodes per class as the training set, leaving the validation and test sets unchanged. Considering each backbone model (i.e., GCN, SGC and MLP), our joint-cluster (JC) loss consistently diliver a much higher node classification accuracy.
>
> | Cora | GCN+CE | GCN+JC | SGC+CE | SGC+JC | MLP+CE | MLP+JC |
> |   :----: |    :---: |    :---: |    :---: | :---: |    :---: | :---: |
> | 5 |  71.80±2.48  |  72.38±2.73 | 72.61±2.59 | 73.67±2.72 | 44.37±3.55 | 51.25±2.76 |
> | 10 |  78.08±1.13 | 78.86±1.45 | 79.34±0.70 | 79.49±1.73 | 53.66±1.55 | 59.19±3.28 |
>
> | CiteSeer | GCN+CE | GCN+JC | SGC+CE | SGC+JC | MLP+CE | MLP+JC |
> |   :----: |    :---: |    :---: |    :---: | :---: |    :---: | :---: |
> | 5 |  57.91±6.18  | 60.98±5.08  | 56.67±5.32 | 61.34±6.56 | 45.52±3.83 | 46.84±5.00 |
> | 10 |  68.22±1.36 | 69.92±1.28 | 68.62±1.13 | 69.88±1.14 | 53.47±2.52 | 55.85±2.22 |
>
> | PubMed | GCN+CE | GCN+JC | SGC+CE | SGC+JC | MLP+CE | MLP+JC |
> |   :----: |    :---: |    :---: |    :---: | :---: |    :---: | :---: |
> | 5 |  70.36±4.79  | 72.00±4.96  | 70.77±3.77 | 71.85±4.65 | 63.45±2.27 | 67.84±3.76 |
> | 10 | 74.94±2.61 | 75.65±1.71 | 75.38±2.83 | 75.79±2.17 | 67.28±1.41 | 71.72±2.14 |
>
> **Q5:** Following the experimental settings of original papers, for GAT, we choose the model parameters by utilizing an early stopping strategy with a patience of 100 epochs on classification cross-entropy loss. For other models(GCN, SGC, MLP and GrapgSAGE), we utilize the model parameters which perform best on the validation set for testing. The remaining hyper-parameters including learning rate, dropout and weight decay are tuned for different models.
>
> **Q6:** In Table 2 of manuscript, we have already demonstrated the advantages on two extremely imbalanced datasets, LastFMAsia and ogbn-arxiv. We attribute this result to the referential ability of joint-cluster distribution modeling, which refers to a cluster of nodes when making decision on target node label during training and inference. The joint distribution weakens the over-confident prediction on the majority classes by assigning prediction confidence on other related minority classes at the cluster, and thus ameliorates the generalization performance. However, in the extremely imbalanced datasets, there are a large number of majority class nodes in the reference cluster of minority class nodes, making the reference information of nodes in the imbalanced datasets not as rich as in the balanced datasets. We think this leads to the improvement on balanced data is higher than imbalanced data.
>
> ---
> Reference:
>
> [1] Thomas N. Kipf and Max Welling. Semi-Supervised Classification with Graph Convolutional Networks. ICLR, 2017.

---

### Official Review · Reviewer_U1a1 · 2023-11-02

**Soundness:** 3 good
**Presentation:** 3 good
**Contribution:** 2 fair
**Rating:** 3
**Confidence:** 3

**Summary:**

This paper argues that existing approaches for graph learning assume independent cross-entropy loss and ignores the inter-dependence induced by observed graph structures. In light of this, the authors propose a new objective that incorporates the inter-dependence among node points into the loss computation. This ensures that the loss for each node is dependent on other nodes during the training. Experiments on many benchmark datasets and using various GNNs as backbones verify the effectiveness of the new loss over the traditional loss function.

**Strengths:**

1. The paper is well written and easy to follow

2. The proposed method seems reasonable and sound

3. Experiments entail a lot of datasets and different GNNs as backbones

**Weaknesses:**

The major concern on this work is the potential over-claiming. The authors argued that existing approaches ignore the inter-dependence among node points for loss computation, which is incorrect. There are in fact quite a few existing works that already considered designing inter-dependent loss for graph learning tasks.

For example, [1] proposes a new objective based on conditional random field for node classification, and [2] harnesses label propagation as a re-weighted loss. Besides, there are also recent works proposing self-supervised loss that considers enforcing the consistency between connected nodes [3]. These approaches integrate the inter-dependence of nodes into the loss function for training.

Another weakness lies in the comparison in experiments. The current experiment only compares with the traditional cross-entropy loss, which is a very weak baseline. More comparison with other advanced methods, particularly the above-mentioned models are needed to well justify the efficacy of the new design.

[1] Meng Qu, et al., Neural Structured Prediction for Inductive Node Classification, ICLR 2022

[2] Hande Dong, et al., On the Equivalence of Decoupled Graph Convolution Network and Label Propagation, WWW 2021

[3] Hengrui Zhang, et al., Localized Contrastive Learning on Graphs

**Questions:**

See weakness above

---

> ### Author Response · Authors · 2023-11-19
> **Response to Reviewer U1a1**
>
> We really appreciate the reviewer for the agreements on our technical contributions and presentation. We would like to address the concerns one by one in the following.
>
> **W1:** Appreciate for the pointing out of these related work. It should be clarified that we are not overemphasizing the contribution of our method, and we will revise the paper to make our contribution clearly understood. We agree that our method only has the same motivation with the above-mentioned models, but this does not overshadow the multifold contributions of this work. The motivation of node labels are dependent is a common sense in graph community (e.g. GNNs utilize message passing mechanism, label propagation[2] distributes labels along edge, CRF-based methods[1] focus on modeling the local label correlation of every linked node pair and contrastive methods[3] treat neighbors as positive samples), but it is more important in how we can incorporate this knowledge.
> Comparing with each mentioned work, we list the detailed differences below:
> * SPN[1] for inductive setting models the local label correlation of each linked node pair via nodeGNN and edgeGNN and takes all edges in whole graph as input to propagate all the pairwise label correlations along edges. In contrast, we train and infer the joint distribution of the target node only with one reference signal (i.e., cluster), which allows the batch training on large graphs (e.g., Amazon with millions of nodes).
> * PTA[2] analyzes the relationship between label propagation and decoupling GNN based on feature propagation. We will supplement this excellent work in the label propagation section of related work. We believe that this method still belongs to the label propagation paradigm, which utilizes label propagation to assign pseudo labels to unlabeled nodes and then performs training.
> * Local_GCL[3] proposes a graph contrastive learning method, which creates positive pairs from first-order neighbors. Different from self-supervised contrastive learning, which measures the feature similarity of a pair of data, instead, we extend to measure the joint label distribution of the target node and its cluster.
>
> We introduce a new paradigm of joint-cluster supervised learning for graph data, which breaks the i.i.d assumption in independent cross-entropy loss computation and model the joint distribution between the target node and its located cluster. This perspective on viewing and handling label dependencies between nodes has never appeared before, and we believe it is novel enough.
>
> **W2:** Thank you for your suggestion. We compare with each mentioned work and list the node classification accuracy on citation networks below. We follow the standard data split. Our method is orthogonal to label propagation and contrastive learning, and they can be effectively combined with our joint-cluster loss, such as using the pseudo-labels assigned by label propagation to make the cluster information richer or introducing node-cluster level contrastive objectives in the representation learning process. We intentionally keep our method simple to illustrate the effectiveness of joint-cluster loss and do not utilize unlabeled nodes during the training process, unlike PTA[2] which assigns pseudo labels to unlabeled nodes and Local_GCL[3] which constrains the representation learning process of unlabeled nodes. It can be found that our method achieves results second only to Local_GCL on small graph data, and delivers the best performance on larger graph arxiv. The SPN[1] requires the edges between labeled nodes for training due to its edge_GNN component. Under the transductive setting, the extremely small number of edges makes it difficult for SPN to model the information of node pairs. For arxiv, SPN cannot be trained on titan RTX24G GPU due to high memory requirements for training each node-pair, which reflects the advantage of utilizing cluster as the reference signal. In addition, our method demonstrates good scalability on millions of large-scale multi-class graphs (Yelp and Amazon), where multi-class tasks are a scenario not covered by the above mentioned works.
>
>
> | Methods | Cora | CiteSeer | PubMed | Arxiv
> |   :----: |    :---: |    :---: |   :---: |   :---: |
> | GCN | 81.70 | 71.43 | 79.06 | 71.74
> | SPN [1] | 80.90 | 69.97 | 76.66 | OOM
> | PTA [2] | 83.55 | 72.89 | 79.52 | 62.79
> | Local-GCL [3] | 84.5 | 73.6 | 82.1 | 71.34
> | GCN_JC | 83.51 | 72.97 | 79.80 | 72.17
>
> ---
> Reference:
>
> [1] Meng Qu, et al., Neural Structured Prediction for Inductive Node Classification, ICLR 2022
>
> [2] Hande Dong, et al., On the Equivalence of Decoupled Graph Convolution Network and Label Propagation, WWW 2021
>
> [3] Hengrui Zhang, et al., Localized Contrastive Learning on Graphs

---

### Meta-Review · Area_Chair_Gpwn · 2023-12-07

**Metareview:**

Paper argues that the average cross entropy loss used for node classification on graphs using GNNs does not take into account the graph structure of the data. As an alternative, the paper proposes joint cluster which also tries to predict the cluster membership of the node in the graph. These clusters are computed using standard graph cluster algorithms. It is shown that the new loss function can improve performance and robustness of some GNNs on classification problems. While this is novel idea with promising results, paper does not provide enough comparison to state-of-the-art GNNs and other competing graph-level loss ideas.

**Justification For Why Not Higher Score:**

Experiments do not compare to prior state of the art results.

**Justification For Why Not Lower Score:**

N/A

---

### Decision · Program_Chairs · 2024-01-16

Reject